# Improving the Utilization of *Flammulina velutipes* Waste during Biochar-Amended Composting: Emphasis on Bacterial Communities

**Longjun Chen** [1,*], **Yu Lin** [2], **Cenwei Liu** [1], **Hui Zhang** [1] and **Chenqiang Lin** [1]

[1] Institute of Resources, Environment and Soil Fertilizer, Fujian Academy of Agricultural Sciences, 247 Wusi Road, Gulou District, Fuzhou 353000, China; liucenwei@faas.cn (C.L.); zhanghui1-tfs@faas.cn (H.Z.); linchenqiang@faas.cn (C.L.)

[2] Freshwater Fisheries Research Institute of Fujian, No. 555 Xihong Road, Gulou District, Fuzhou 353000, China; linyuuu@126.com

* Correspondence: chenlongjun@faas.cn

**Abstract:** This study investigated the impacts of biochar addition on N conversion, humification, and bacterial community during *Flammulina velutipes* waste composting. The mixture of chicken manure and *Flammulina velutipes* waste was 4:6 (dry weight basis). The biochar was added into the mixture and mixed thoroughly at ratios of 0, 2.5, 5, and 7.5% (*w/w*) and labeled as CK, T1, T2, and T3, respectively. The results showed that the biochar treatment significantly improved the compost maturity by increasing humic substances and the conversion of $NH_4^+$-N to $NO_3^-$-N. With the increase in biochar supplemental level, the abundance, diversity, and uniformity of the microbial community were improved. The dominant taxa were *Firmicutes*, *Bacteroidota*, *Actinobacteriota*, *Proteobacteria*, and *Gemmatimonadota*, especially the *Firmicutes* and *Bacteroidota*. Biochar addition facilitated the proliferation of thermophilic bacteria such as *Bacillus*, *Actinobacteriota*, *Parapedobacter*, and *Sphingobacterium*, leading to enhanced organic decomposition to increase humus. The findings of this study highlighted the positive effects of biochar addition on the composting mixture of chicken manure and *Flammulina velutipes* waste. These results can help to produce high-quality biochar composting products by balancing organic decomposition and humification based on the bacterial community.

**Keywords:** *Flammulina velutipes* waste composting; biochar; bacterial community; N conversion; humification

## 1. Introduction

In China, various types of mycelium residues (about 22.38 million tons a year) are generated as 25~33% of fresh mushroom production [1]. *Flammulina velutipes* waste is the residue after collecting the fruiting bodies of mushrooms. Currently, the utilization of mushroom waste is used as animal feed, composting, or polysaccharide isolates [1–4]. According to previous research, the production of mushrooms has increased globally, resulting in approximately 53 million tons of mushroom waste being produced, calculated on the basis of 1 kg of mushrooms requiring 5 kg of media substrate [3]. As mushroom production continues to increase, a new application for mushroom waste or spent mushroom substrate is necessary. An important part of the mushroom growing process is the fruiting body of the mushrooms, which is frequently disposed of as waste [5]. It was converted into flour and utilized as cookies, steamed buns, and mushroom-chicken patties [5]. Other research has shown that the mushroom waste could regulate soil nutrition as a soil amendment, improving plant growth [6]. Thus, finding a rational way to use mushroom waste is important.

For agricultural production, chemical fertilizers or animal waste are inevitably used. Livestock waste was used as an important organic fertilizer raw material based on valuable

nutrients, especially nitrogen [7,8]. It is worth noting that animal waste is an important source of pathogenic microorganisms such as *Salmonella*, *Escherichia coli*, *Staphylococcus*, *Streptococcus*, *Clostridium*, *Listeria*, *Campylobacter*, *Corynebacterium*, and *Mycobacterium* [9]. Although chicken manure contains many pathogens, it is also used as a fertilizer for crops after being reasonably treated, including incineration, anaerobic digestion, direct burning, and composting [6–8,10,11]. Currently, in China, poultry production is growing rapidly based on the excellent low cholesterol properties of poultry meat, and it provides about 82% of meat production [12]. However, the massive poultry production results in a huge amount of excrement, which is considered a major source of poultry waste [7]. At present, increasing applications of organic fertilizer is an effective way to improve crop yield [13–15]. Previous studies have shown that organic fertilizers can optimize rhizosphere bacterial community structure and improve soil fertility and soil continuous production capacity [13,16]. On the one hand, organic fertilizer can provide more adequate nutrition, regulate and stimulate crops, and control and reduce the harm of diseases and insects in the process of crop growth [17]. On the other hand, organic fertilizers can reduce soil compactness and improve soil porosity, which is conducive to the utilization of nutrients [14].

Biochar as a conventional composting material is environmentally friendly, resulting in cost effectiveness, promoting the availability and distribution of food, and facilitating planetary conservation [18]. Another purpose, using biochar in composting, could reduce pathogenic microorganisms and improve plant nutrients [19]. It is well known that chicken manure is an important source of pathogenic microorganisms such as *Salmonella*, *Escherichia coli*, *Staphylococcus*, *Streptococcus*, *Clostridium*, *Listeria*, *Campylobacter*, *Corynebacterium*, and *Mycobacterium* [9]. Although chicken manure contains many pathogens, it can be used effectively as a fertilizer and is useful for incineration, anaerobic digestion, direct burning, and compost [7,10,11,20,21]. Importantly, composting can enhance the soil environment by increasing soil organic matter, nitrogen, phosphorus, and many trace elements [22,23]. In addition, compound organic fertilizer with biochar can increase the soil organic carbon content and the activity of acid phosphatase, catalase, and other microbial enzymes [21,24,25].

At present, composting is the ideal environmentally friendly technology, using microorganisms to convert organic waste into humus-like substances and then obtain organic fertilizer and biomass resources [22]. According to previous research, in in-vessel composting, biochar as an additive was an important carbonaceous material [26]. Biochar could also stimulate microbial activity to improve the quality of the compost product, which was proven by comparing it to other materials have been widely investigated such as sheep manure, pig manure, and chicken manure [7,26–28]. However, information about the effects of biochar on the composting quality of mushroom waste and the microbial response was limited. In this study, we hypothesized that the addition of biochar to the mixture of mushroom waste and manure could improve the quality of the compost products and shift the microbial community. Therefore, the aims of this study were: (1) to investigate the effects of different ratios of biochar on the main parameters of the composting products of mushroom waste and livestock manure; (2) to explore the microbial response of the composting products.

## 2. Materials and Methods

### 2.1. Compost Treatment, Raw Materials and Sample Collection

The raw materials of chicken manure, *Flammulina velutipes* waste, and biochar were used in the composting. The medium of *Flammulina velutipes* provided by Fuzhou Yida Food Co., Ltd. (Lianjiang, Fuzhou, China). It mainly consisted of corn cobs, rice bran, bran, corn meal, and cottonseed shells and after harvesting, and some of the *Flammulina velutipes* mycelium waste was separated. The fresh chicken manure was collected from a large-scale livestock farm located in Fujian Sunner Development Inc. (Nanping, China). Biochar (corn stalk, grain size 100 mesh, specific surface area 1000–1300 m$^2$/g, C content 95%, ash 5%) was purchased from Pingdingshan Lvzhiyuan Activated Carbon Co., Ltd.

(Pingdingshan, China). The composting process was conducted in the compost research laboratory of the Fujian Academy of Agricultural Sciences. The composting process referred to Ravindran's method, using a 60 cm × 60 cm × 55 cm container [21]. The ventilation mode was intermittent, the fan was started for 2 min every 60 min, and the ventilation rate was 0.2 L/(kg·min). The mixture of chicken manure and *Flammulina velutipes* waste was 4:6 (dry-weight basis). The biochar was added into the mixture and mixed thoroughly at ratios of 0, 2.5, 5, and 7.5% (*w/w*) and labeled as CK, T1, T2, and T3, respectively. Additionally, the temperature of the compost and the ambient temperature were measured daily at 9:00 am and 5:00 pm. The compost was refurbished weekly, then about 500 g of sample was collected on days 0, 7, 14, 21, 28, 35, 42, 49, and 77. The samples were divided into two parts. One part was stored at −4 °C for determination, and the other part was dried and crushed for reserve use.

### 2.2. Determination Items and Methods

Air-dried samples were used for the determination of electrical conductivity (EC), measured in a 1:5 (*w/v*) aqueous extract using a conductivity meter (DDSJ-308A, Shanghai, China). The pH was determined from soil–water suspensions (1:10 *w/v*) using a pH meter (AB150, Thermo Fisher Scientific Inc., Beijing, China). A 20 g soil sample was extracted using 100 mL of ultrapure water and filtered through 0.45 membrane to analyze the dissolved organic carbon (DOC) and total nitrogen (TN). The total organic carbon (TOC) and TN was determined using an elemental analyzer of Liqui TOC II (Elemetar Liqui TOC, Elementar Co., Ltd., Hanau, Germany) and a UV-1700 Pharma Spec visible spectrophotometer (220 nm and 275 nm), respectively. The germination index (GI) was determined as reported in references [29,30]. Chinese cabbage seeds were used and germinated on a tray to analyze the GI.

According to previous reports [31,32], the humification was quantified by determining the humus content. The humic acid (HA) and fulvic acid (FA) were separated from the humus using centrifugation at 4400 rpm with mixture of 1 g of sample, 0.1 M sodium pyrophosphate ($Na_4P_2O_7 \cdot 10H_2O$), and 0.1 M NaOH. The FA in the supernatant and the HA in the precipitate were collected, then the HA was extracted from the precipitate sample by re-dissolving it in 0.1 M NaOH. Finally, the FA and HA contents of the humus were quantified using a total organic carbon (TOC) analyzer (TOC-L, Fuzhou, China). The humification properties of the composting products were determined using ultraviolet–visible (UV–vis) spectroscopy (N4S, Huachen, China) according to a previous method [31]. The SUVA280 and E4/E6 ratio was determined base on the ratio of absorbance at 280, 465, and 665 nm, respectively. Finally, the humus aromaticity and molecular weight of the compost were calculated [31]. The soil inorganic N in 20 g of fresh soil samples was extracted using 100 mL of $K_2SO_4$, filtered through a 0.45 μm membrane and analyzed using a LACHAT Quikchem Automated Ion Analyzer (Hach Corp, Loveland, CO, USA). A more detailed description of the measurements of the soil $NH_4^+$-N and $NO_3^-$-N can be found in Li et al. [8] and Fu et al. [33].

### 2.3. DNA Extraction and High-Throughput Sequencing

DNA extraction was performed using an Omega DNA Kit (Omega Bio-Tek, Norcross, GA, USA) following the manufacturer's instructions. Qualitative PCR was performed according to the instructions contained in the Fast DNA® Spin Kit for Soil (MP Biomedicals, Irvine City, CA, USA) based on the DNA production evaluated using a NanoDrop spectrophotometer (ND2000; Thermo Scientific, Waltham, MA, USA) and agarose gel electrophoresis, respectively. The V3-V4 hypervariable region of 16S rRNA genes was amplified using the primers 343F TACGGRAGGCAGCAG and 798R AGGGTATCTAATCCT equipped with 12-base barcodes for sample distinction. A 25 μL reaction solution containing 12.5 μL of PCR MasterMix, 5 μM of each primer, 5.5 μL of ddH₂O, and 10 ng of template DNA was used for the analysis. The PCR conditions were set to 98 °C for 1 min, followed by 35 cycles for denaturation at 98 °C for 20 s, annealing at 50 °C for 60 s, extension at 72 °C for 30 s, and a final extension at

72 °C for 5 min. The PCR product was extracted and purified using a Qiagen DNA gel extraction kit (Qiagen, Valencia, CA, USA) and 1.5% agarose gels. A sequencing library was generated using a TruSeq® DNA PCR-Free Sample Preparation Kit (Illumina, San Diego, CA, USA). The library was applied to an Illumina NovaSeq PE250 platform by Oebiotech Bio-Pharm Technology Co. Ltd. (Shanghai, China).

### 2.4. Bioinformatic and Statistical Analysis

A one-way analysis of variance (ANOVA) and Duncan multiple comparisons test were used to analyze the high-throughput sequencing data using SPSS (version 21.0, IBM, New York, NY, USA) ($p < 0.05$). Visualization of the OTUs for each sample was performed using the online platform Genes Cloud (https://www.genescloud.cn, accessed on 11 January 2024). An LEfSe analysis and the LDA scores were completed using the Wekemo Bioincloud platform (https://www.bioincloud.tech, accessed on 11 January 2024). The data are reported as means and standard deviations (SDs). Statistical significance was set at $p < 0.05$.

## 3. Results

### 3.1. Effects of Biochar on Physiochemical Properties and Humification during Composting

The temperature, pH, EC, and GI of the composting products were characterized. The temperature in all the treatments broadly followed a similar profile, including the four critical mesophilic, thermophilic, cooling, and maturation steps (Figure 1A). The temperature of the compost first increased and then decreased. After each artificial turning, the temperature rose slightly and eventually settled at to room temperature (Rt). The results demonstrated that the biochar addition significantly increased the temperature compared with the control group during the thermophilic period during composting ($p < 0.05$), and the duration of the high-temperature period (above 50 °C) was more than 8d.

The matrix pH increased rapidly in the initial stage of composting and then stabilized in the mesophilic stage for all the treatments (Figure 1B). The biochar addition could obviously reduce the EC value of the compost products compared with the CK ($p < 0.05$, Figure 1C). According to the GI (Figure 1D), the biochar addition was more notable for the compost compared with the control treatment. The GI ranged from 0.178 to 2.01 in all the groups. The biochar treatment increased the GI after 20 days of composting. The GI was greater than 50% after 20 days of composting, indicating that the compost had reached maturity.

In Figure 1E, the humus content in all the treatments increased incessantly throughout composting. Increased humus is associated with changes in HA and FA. The two results were opposite (Figure 1F,G). In the initial stage of composting, a large amount of organic matter was decomposed, and fulvic acid, which is easier to decompose, was fully utilized in the high-temperature stage of composting (Figure 1A). The FA content was obviously decreased in the initial stage of composting (Figure 1G). When the biochar was added, there was an apparent increase in the humus and HA contents (Figure 1E,F) compared with the CK, but there was an apparent reduction in the FA contents in the thermophilic stage of composting. When the compost entered the cooling period (during d25–d40 period), the HI value had a slight downward trend (Figure 1H). In order to determine the alteration on humification, the $SUVA_{280}$ and $E_4/E_6$ were also determined. The results showed that a high $SUVA_{280}$ value (Figure 1I) and low $E_4/E_6$ value (Figure 1J) indicated a high degree of maturity and stabilization. In this study, the $SUVA_{280}$ value and $E_4/E_6$ values of the composting products increased by 14.78% or decreased by 8.75% with the increases in the biochar addition, respectively.

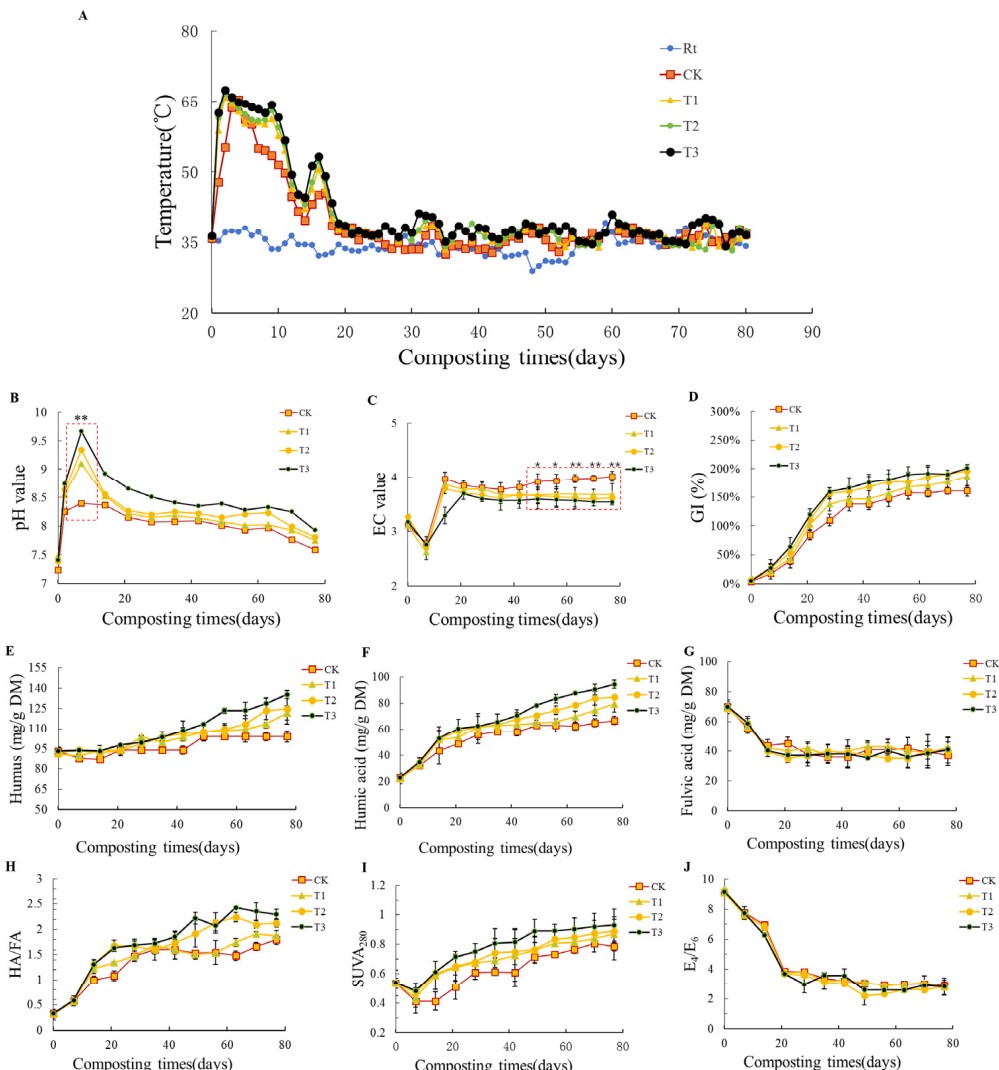

**Figure 1.** Effects of biochar addition on (**A**) temperature, (**B**) matrix pH, (**C**) electrical conductivity (EC), (**D**) germination index (GI), (**E**) total humus, (**F**) humic acid (HA), (**G**) fulvic acid (FA), (**H**) HA/FA, (**I**) SUVA$_{280}$, and (**J**) E$_4$/E$_6$ during composting. CK, control, without biochar addition; T1, 2.5% biochar treatment; T2, 5% biochar treatment; T3, 7.5% biochar treatment. * means $p < 0.05$; ** means $p < 0.01$.

### 3.2. Effects of Biochar Addition on C and N Conversion

The changes in the organic carbon can reflect the degree of compost maturation to a certain extent. The TOC was decreased during the composting process (Figure 2A). In the early stage of composting, the compost pile contained more easily decomposed organic matter, and the mass propagation of microorganisms caused the rapid decline in organic carbon. Different from TOC, the DOC can be directly decomposed and utilized by microorganisms in the compost mass, and it is an important indicator that is used to reflect the microbial activities and rate of composting. According to Figure 2B, the DOC content was obviously reduced with the 7.5% biochar treatment ($p < 0.05$). The biochar addition could increase the N conversion (Figure 2C–E). The content of TN of the final compost product was decreased with the biochar addition (Figure 2C). The biochar addition significantly decreased the content of NH$_4^+$-N (Figure 2D) and increased the content of NO$_3^-$-N (Figure 2E). According to the red arrow in Figure 2E,F, the conversion of NH$_4^+$-N to NO$_3^-$-N in the compost products showed a one-to-one correspondence at different sampling times. The C/N ratio was also higher in the biochar addition group than in the control group (Figure 2E), especially in the initial period of composting.

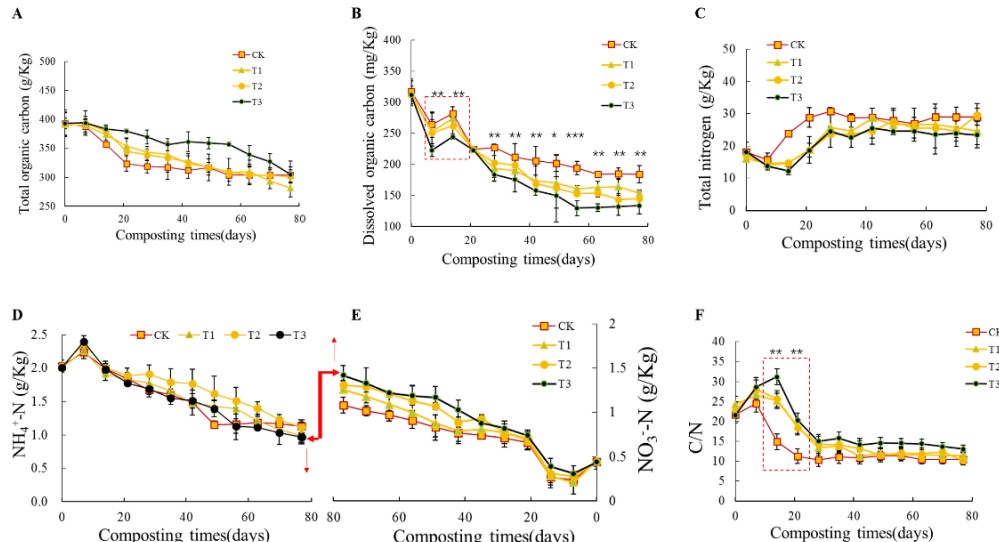

**Figure 2.** Effects of biochar addition on C and N conversion during the composting process. (**A**) total organic carbon (TC), (**B**) dissolved organic carbon (DOC), (**C**) total nitrogen (TN), (**D**) $NH_4^+$-N, (**E**) $NO_3^-$-N, (**F**) C/N ratio. The red arrow showed that the conversion of $NH4^+$-N to $NO3^-$-N in compost products at different sampling times. CK, control, without biochar addition; T1, 2.5% biochar treatment; T2, 5% biochar treatment; T3, 7.5% biochar treatment. * means $p < 0.05$; ** means $p < 0.01$; *** means $p < 0.001$.

### 3.3. Effects of Biochar on Bacterial Community Diversity and Composition

The biochar treatment was found to increase the bacterial community's diversity during composting (Table 1). The α-diversity indices decreased notably and then increased in all the group with biochar addition. These results indicated that the bacterial community compositions during composting were different among the different biochar treatments. The differences in the bacterial community compositions were also reflected in the relative abundances of bacterial taxa at the phylum level (Figure 3A,C). The dominant taxa were *Firmicutes*, *Bacteroidota*, *Actinobacteriota*, *Proteobacteria*, and *Gemmatimonadota*, with a total abundance of above 95%. It is worth noting that the biochar addition obviously increased the *Nitrospirota* content (Figure 3A), which belongs to the *Nitrobacter* genus. Furthermore, *Firmicutes* was the most abundant, with a notable increase in the relative abundance with the biochar treatment (Figure 3A). The heat map in Figure 3B depicts the relative abundances of the top 15 genera, with the dominant genera being *Galbibacter* in the T3 group. This result is consistent with the changes in temperature (Figure 1A). At the mature stage of composting, the *Pseudomonas*, *Flavobacterium*, *MWH-CFBk5*, *Parapedobacter*, and *Sphingobacterium* were key contributors to the phylum *Bacteroidetes* for the all the treatment groups (Figure 3B).

**Table 1.** Alterations in α-diversity indices during composting.

| Treatment | Time (d) | Chao1 | Shannon | Observed Species | Simpson | PD Whole Tree | ACE |
|---|---|---|---|---|---|---|---|
| CK | 0 | 789 | 6.66 | 783 | 0.97 | 71 | 787 |
| | 7 | 466 | 6.45 | 462 | 0.97 | 39 | 465 |
| | 21 | 533 | 6.43 | 526 | 0.96 | 46 | 529 |
| | 35 | 493 | 6.56 | 487 | 0.97 | 44 | 494 |
| | 77 | 527 | 6.65 | 521 | 0.97 | 47 | 527 |
| T1 | 0 | 661 | 6.28 | 657 | 0.96 | 61 | 660 |
| | 7 | 423 | 6.25 | 419 | 0.97 | 35 | 418 |
| | 21 | 467 | 6.40 | 463 | 0.96 | 41 | 468 |
| | 35 | 586 | 6.88 | 581 | 0.98 | 50 | 585 |
| | 77 | 464 | 6.60 | 460 | 0.97 | 41 | 464 |

**Table 1.** *Cont.*

| Treatment | Time (d) | Chao1 | Shannon | Observed Species | Simpson | PD Whole Tree | ACE |
|---|---|---|---|---|---|---|---|
| T2 | 0 | 806 | 7.24 | 801 | 0.98 | 70 | 804 |
|  | 7 | 498 | 6.41 | 492 | 0.97 | 41 | 496 |
|  | 21 | 480 | 6.47 | 474 | 0.97 | 42 | 474 |
|  | 35 | 441 | 6.45 | 437 | 0.97 | 39 | 437 |
|  | 77 | 698 | 6.99 | 694 | 0.98 | 59 | 696 |
| T3 | 0 | 565 | 6.44 | 562 | 0.97 | 55 | 563 |
|  | 7 | 658 | 6.30 | 654 | 0.95 | 55 | 656 |
|  | 21 | 473 | 6.18 | 468 | 0.96 | 42 | 470 |
|  | 35 | 752 | 6.76 | 745 | 0.97 | 61 | 747 |
|  | 77 | 583 | 6.31 | 578 | 0.97 | 50 | 579 |

Note: CK, control, without biochar addition; T1, 2.5% biochar treatment; T2, 5% biochar treatment; T3, 7.5% biochar treatment; PD, phylogenetic diversity; ACE, abundance-based coverage estimator metric.

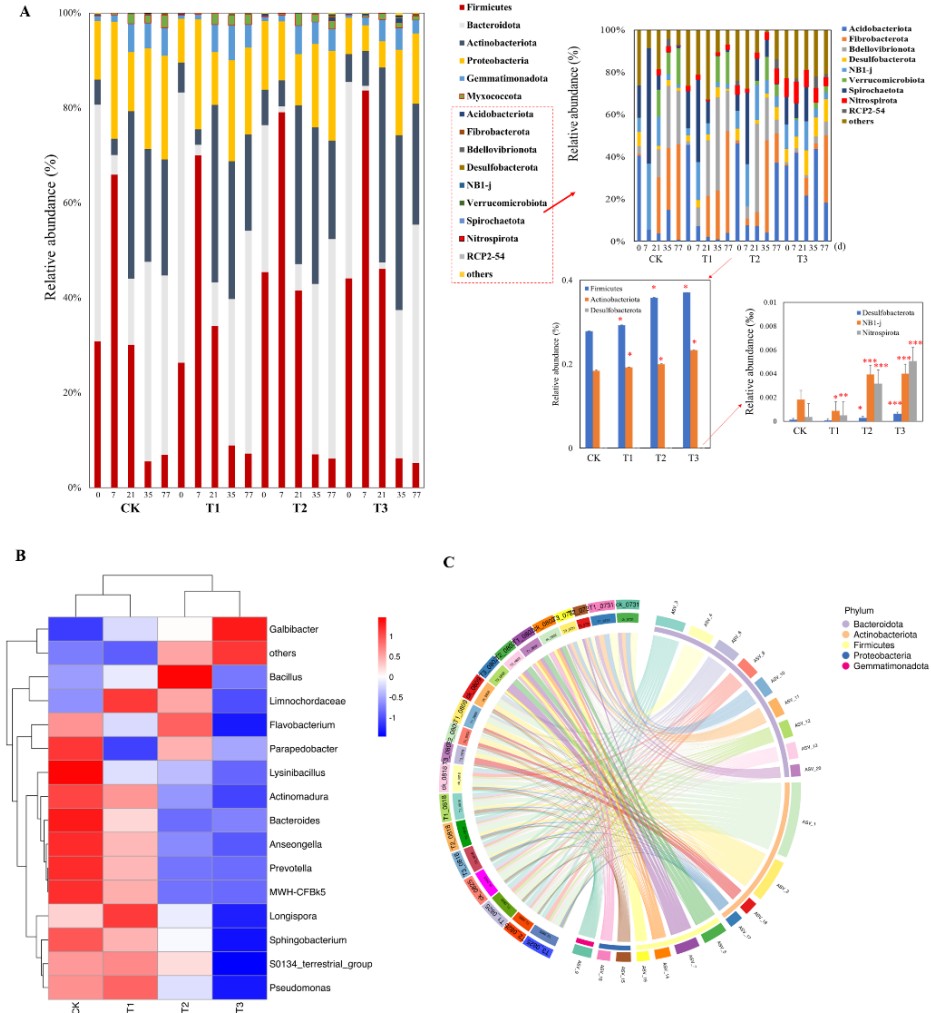

**Figure 3.** Relative abundance of the predominant bacterial community at the phylum level (**A**), genus levels (**B**), and circos at phylum (**C**) during composting. "Others" have a relative abundance of <1%. * means $p < 0.05$; ** means $p < 0.01$; *** means $p < 0.001$.

### 3.4. Relationship between Bacterial Community and Composting Properties

A network analysis was used to identify the correlation among the bacteria, C-N conversion, and humification properties (Figure 4). As shown in Figure 4A,C, the C-N conversion had a significant correlation with several abundant bacteria. Especially, the TOC and C/N contents exhibited significantly positive correlations with the genera *Bacillus*. This genus was abun-

dant at the thermophilic stage of composting to decompose the organic compounds, thereby inducing considerable carbon loss (low C/N ration and TOC, as shown in Figure 2A,C). Moreover, the genera *Prevotella*, *Bacteroides*, and *Fastidiosipila* were significantly negatively correlative with TN and $NO_3^-$-N, whereas the $NH_4^+$-N had a significantly positive correlation with the genus *Prevotella*, indicating that the organic substances were rapidly decomposed at the thermophilic stage of composting (Figure 4A,C). As shown in Figure 4B,D, the HA, HI, SUVA$_{280}$, and total humus were positively correlated with the genera *Flavobacterium*, *MWH-FBk5*, *Parapedobacter*, and *Sphingobacterium* in all the treatment groups. These results indicate that the biochar addition enhanced the formation of functional groups associated with a high abundance of the phylum *Bacteroidetes* in composting.

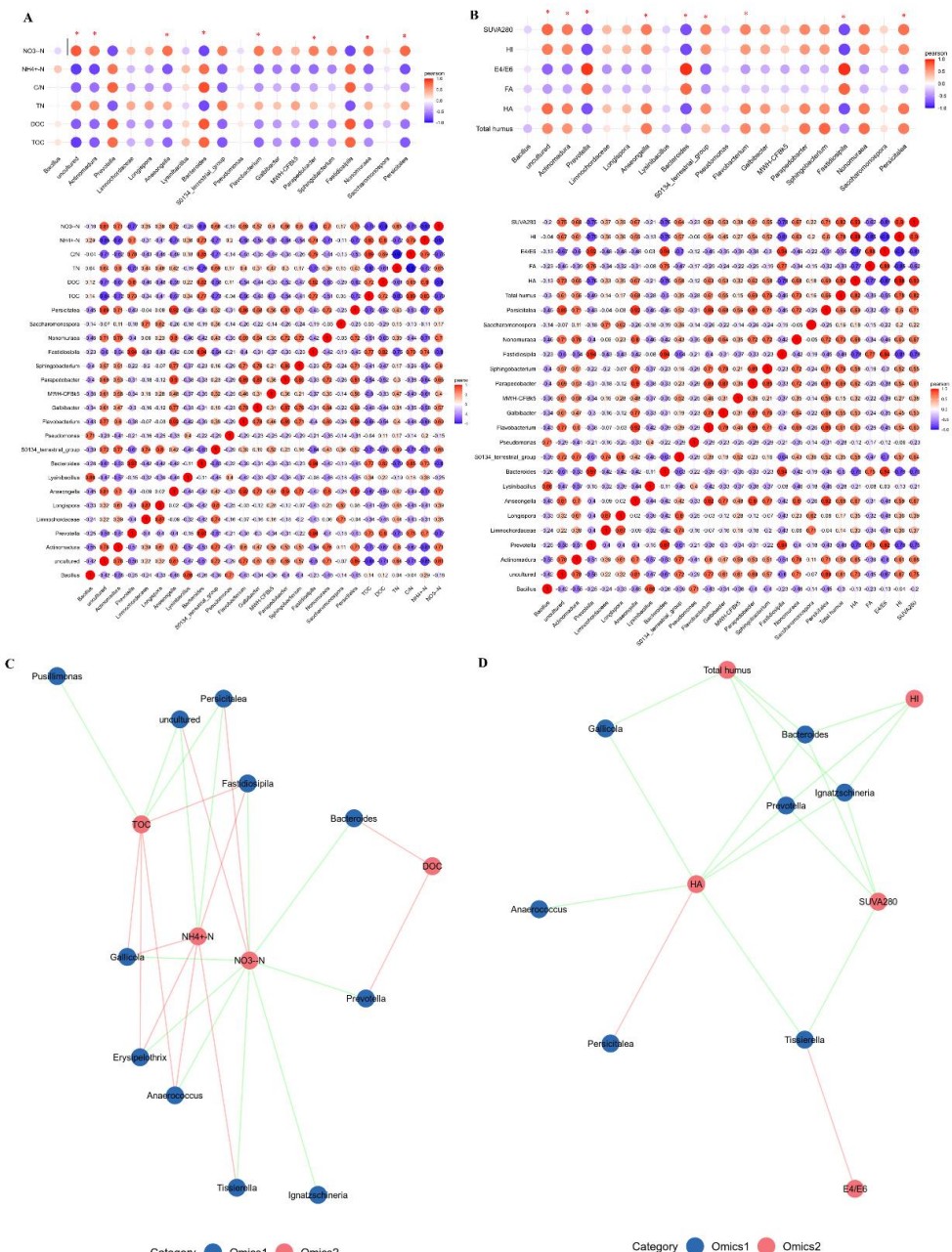

**Figure 4.** Relationships between bacterial community and C and N conversion or humification properties based on Pearson results (**A**,**B**) and Spearman results (**C**,**D**). Red and blue colors represent positive and negative correlations in the Pearson results, respectively. Red and green lines represent positive and negative correlations in the Spearman results, respectively. Pink and blue nodes represent C-N conversion and humification, respectively, with the size indicating the connectivity. * means *p* < 0.05.

The composting bacterial function with the biochar treatment was analyzed using PICRUSt based on the Kyoto Encyclopedia of Genes and Genomes (KEGG) database (Figure 5). In the present study, the relative abundance of six function categories of level-1 was recorded (Figure 5A). Metabolism was the primary factor, with a relative abundance of more than 90%, and the rest were less than 6%, including the ABC transporter and carbon fixation in photosynthetic organisms. The result showed that the biochar treatment, especially at a rate of 7.5% (T3 group), increased the relative abundance of some bacterial genes associated with amino acid metabolism (e.g., phenylalanine, tyrosine, tryptophan, and pyruvate metabolism) and carbohydrate metabolism (e.g., fructose and mannose metabolism, galactose metabolism, glycolysis/gluconeogenesis, citrate cycle). The carbon metabolism and nitrogen metabolism were obviously increased in the T3 group compared with the CK ($p < 0.05$). During the composting, the total abundance of functional bacteria for carbon metabolism was increased. Based on the above results, the carbon metabolism was driven by a possible benefit from chemoheterotrophs for all the treatments, such as *Nitrospirota*. In all the treatments, the biochar addition resulted the highest abundance of bacterial *Nitrospirota* (Figure 3A). The genera *Bacillus*, *Actinobacteriota*, *Parapedobacter*, and *Sphingobacterium* were identified as biomarkers of the biochar treatment at the mesophilic and mature stages based on the LEfSe analysis (Figure 5B).

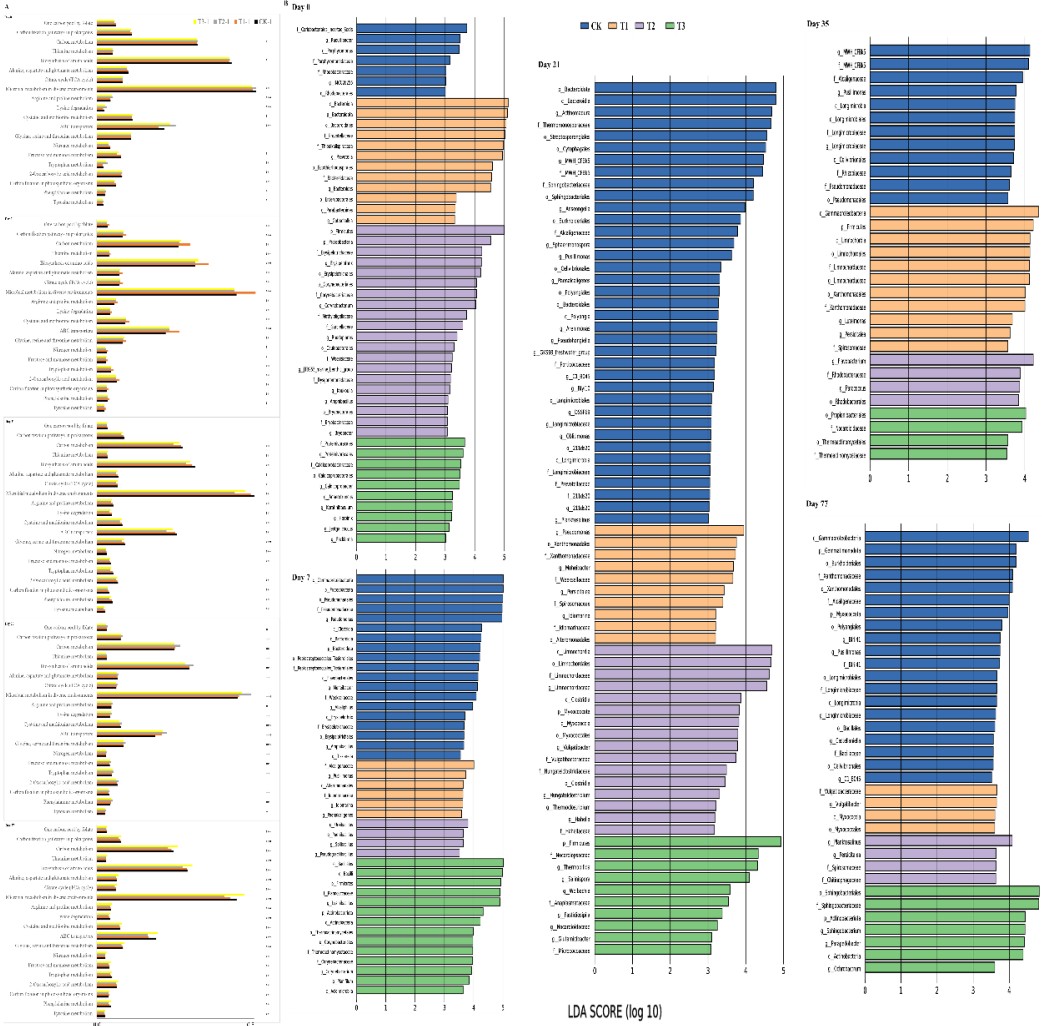

**Figure 5.** Effects of biochar on biomarkers during the composting of a mixture of chicken manure and *Flammulina velutipes* waste. (**A**) C and N metabolism; (**B**) LDA scores of biomarkers in different treatments based on an LEfSe analysis during composting. Gene families are colored by functional categories. CK, control, without biochar addition; T1, 2.5% biochar treatment; T2, 5% biochar treatment; T3, 7.5% biochar treatment. * means $p < 0.05$; ** means $p < 0.01$; *** means $p < 0.001$.

## 4. Discussion

At the initial stage of composting, the compost contains a large amount of organic matter, resulting in bacteria proliferation, which in turn releases a large amount of heat and eventually causes the temperature of the compost to rise [34]. The temperature in the composting piles is increased by enhancing the biodegradation of organic substances [31]. This change in temperature also meets the requirements of the harmless treatment of livestock and poultry waste. Moreover, biochar addition prolongs the high-temperature period of compost, which is attributed to filling the pores of the compost raw materials and reducing the heat loss of the compost pile [35]. On the other hand, the huge specific surface area and pore structure of biochar can provide favorable space conditions for microbial activities and improve microbial activity [8].

At the beginning of the composting process, the pH value increased rapidly, which could be attributed to the biochar treatment increasing the consumption of organic acids. Subsequently, the raw materials of poultry manure were decomposed and produced organic acids, which caused the pH values to decrease gradually [28]. In this study, the pH values of the final product were more than 7.5, which were attributed to the alkaline action of the biochar. Additionally, biochar addition can enhance ammonification in composting to rapidly reduce organic acid contents [31]. The EC is an important indicator of the soluble salt content of reactive compost products and also indicates the potential phytotoxicity of the composting products [27]. Biochar has a strong adsorption on water-soluble ions, resulting in a decrease in the soluble salt content, which can also be caused by the dilution of biochar on the soluble salt in the compost piles [27]. In the initial stage of composting, a large amount of organic matter is decomposed, and fulvic acid, which is easier to decompose, is fully utilized in the high-temperature stage of composting [36,37]. This alteration is possibly due to the changes in the exogenous microbe. When the compost enters the cooling period, the HI value can decrease, which is attributed to the microorganisms using cellulose, lignin, and other difficult-to-decompose organic matter to form part of the FA content [37]. In order to determine the alterations in humification, the $SUFA_{280}$ and $E_4/E_6$ values were also determined, which showed a high $SUFA_{280}$ value and low $E_4/E_6$ value. This which indicated that the biochar could improve the compost maturity by increasing humic substances [36]. Thus, biochar treatment could decompose the waste and improve the TOC.

The change in organic carbon can reflect the degree of compost maturation to a certain extent; a reduction in the DOC content in the biochar addition was reported during pig manure composting [27]. The biochar addition could also enhance the DOC decomposition by decreasing the composting density [35]. Humus formation involves a complex process of organic decomposition and polymerization to secure compost quality. Previous research has explored inoculating exogenous microbes to accelerate organic decomposition and humification. For example, inoculation of *Bacillus subtilis* to cow manure could enhance the formation of stable humus-like substances [38]. Adding self-cultured Bacillus megaterium could enhance ammonia oxidation and regulate nitrification [39]. Their functions or roles in humification during composting are also largely unknown. Thus, no matter what method is chosen to regulate humification, it is good to change the reuse of waste. Another important indicator for the advantage of compost is the GI. In the present research, the GI was greater than 50% after composting for 20 days. These results are also consistent with previous studies. When the GI value is greater than 50%, it is generally believed that the compost is in the formation stage. When the GI value is greater than 80%, it is completely mature [40].

In the present study, the biochar addition improved the N cycling, which was in accordance with previous studies [14,28,41,42]. The previous study showed that $NO_3^-$-N is a better form of N for many plants than $NH_4^+$-N [43]. A high nitrate concentration in compost is normally desired to improve the compost quality. In compost products, the conversion of $NH_4^+$-N to $NO_3^-$-N showed a one-to-one correspondence, which indicated that the biochar addition was conducive to the formation of $NO_3^-$-N and effectively reduced the loss of N [8]. In previous research, a low C/N ratio could inhibit the utilization

of carbon sources, resulting in a reduced quality of the compost products [7]. However, the content of TN of the final compost product was decreased with the biochar addition. This result was different from previous research showing that biochar addition could obviously increase the TN contents in the final compost product [42]. This difference may be related to the original C/N of the compost. This variation in the difference may be related to raw material use, which we will also verify in further studies.

In previous research, biochar was found to provide favorable space conditions to improve microbial activity [8]. A positive effect of biochar addition on bacterial diversity was reported in rice straw compost with chicken manure mixed with peanut straw compost [8,44]. It has also been reported that some of these phyla were the most predominant in biochar-amended compost. According to the previous results, *Proteobacteria* and *Acidobacteria* are important phyla [31]. *Proteobacteria* played a vital role during the process of nitrogen and carbon cycling and directly affected the quality of the compost [28,45]. *Acidobacteria* represents an underrepresented soil bacterial phylum whose members are pervasive and copiously distributed across nearly all ecosystems. However, *Acidobacteria* possesses an inventory of genes involved in diverse metabolic pathways, as evidenced by their pan-genomic profiles [46]. It is worth noting that the biochar addition obviously increased the *Nitrospirota* (Figure 3), which belongs to the *Nitrobacter* genus. The *Nitrobacter* genus is frequently used as a model strain to drive the nitrogen cycle process by converting ammonia nitrogen into nitrite and nitrate to reduce harmful products [47]. Thus, biochar addition increased the *Nitrospirota* abundance could improve the compost preponderance. *Actinomyces* are also important because they can degrade lignocellulose and secrete multiple antibiotics to eliminate pathogenic microorganisms [48]. At the genera level, *bacteroides* includes *Clostridium* and *Enterobacter*, which have proved to directly promote plant *Ageratina Adenophora* [49]. Furthermore, the dominant genera *Galbibacter* in the T3 group contains an important bacterium in regulating innocent treatment efficiency [50]. As a previous study, the relative abundance of thermophilic bacteria and *Galbibacter* were increased during composting [51]. At the mature stage of composting, the *Pseudomonas*, *Flavobacterium*, *MWH-CFBk5*, *Parapedobacter*, and *Sphingobacterium* were key contributors to the *Bacteroidete* phyla, which were involved in the humification of the organic matter during composting. For example, *Flavobacterium* has been widely used in the degradation of macromolecular organic matter such as lignocellulose [36]. The relative abundance of these bacteria genera was increased, which could raise the contents which from lignocellulose decomposition to humic substances. Thus, this may also be one of the reasons why biochar addition can improve the quality of composting.

The PICRUSt results showed that the metabolism was the most primary based on the KEGG database, especially some bacteria associated with amino acid metabolism and carbon metabolism such as *Nitrospirota*. *Nitrospirota* could rapidly regulate the matrix conditions and increase the carbon fixation rate [52]. According to previous results, biochar addition could increase the amino acid metabolism and carbohydrate mentalism of the bacteria [28], and it also increased the humic substance contents [53]. Importantly, *Bacillus*, *Actinobacteriota*, *Parapedobacter*, and *Sphingobacterium* were identified as biomarkers, which have proved to be beneficial to increasing the C-N conversion and the formation of humic substances [53]. However, the properties of biochar are affected by high moisture, temperature, and organic matter contents [54], especially limiting its sorption capacity. The high potential of compost is associated with the non-carbonized organic matter content and the conditions of the production of biochar [55]. For example, biochar produced at 300 °C has a larger sorption capacity, retains the heavy metals present in the soil on its surface, and reduces the carbon dioxide emissions [56,57]. Therefore, the selection of biochar produced at the right temperature and the addition of the appropriate proportion are conducive to the use of compost.

## 5. Conclusions

Biochar addition significantly enhanced compost maturity and influenced some of the physicochemical properties (temperature, pH, EC, GI, TOC, TN, $NO_3^-$-N, and $NH_4^+$-N) of the final compost product, especially the $NO_3^-$-N content. Biochar addition significantly affected the bacterial community composition, especially the relative abundance of some beneficial taxa. The predominant bacterial taxa included *Bacillus*, *Actinobacteriota*, *Parapedobacter*, and *Sphingobacterium*. The findings of this study highlight the positive effects of biochar addition on the composting mixture of chicken manure and *Flammulina velutipes* waste. These results can help us to produce high-quality biochar composting products by balancing organic decomposition and humification.

**Author Contributions:** L.C.: experimental design, performed the experiment, data curation and analysis, writing—original draft preparation. Y.L. and C.L. (Cenwei Liu): performed the experiment, writing—editing. H.Z. and C.L. (Chenqiang Lin): sample pretreatment, DNA extraction. All authors have read and agreed to the published version of the manuscript.

**Funding:** The authors would like to acknowledge the financial support of the Fujian Province Public Welfare Scientific Research Program (No.: 2022R1025005), Fujian Spark Project (No.: 2023S0064), special fund project for high-quality development of marine services and fisheries in Fujian province (No.: FJHY-YYKJ-2024-2-2), and Explore Scientific and Technological Innovation Projects of the Fujian Academy of Agricultural Sciences (No. ZYTS202414).

**Data Availability Statement:** The data that support the findings of this study are available from the corresponding author upon reasonable request.

**Acknowledgments:** We would like to acknowledge our colleague Alex Zhang, who corrected the manuscript.

**Conflicts of Interest:** The authors declare no conflicts of interest.

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
