# Peer review of "Improving the Utilization of Flammulina velutipes Waste during Biochar-Amended Composting: Emphasis on Bacterial Communities"

_agronomy, doi:10.3390/agronomy14051046_

Round 1

Reviewer 1 Report

Comments and Suggestions for Authors

Composting is one of the most widely accepted circular economy approach for recycling organic waste and represents a cost-effective method for recycling nutrients. Co-composting enables the aerobic degradation of organic waste mixtures to obtain compost that can be used as fertilizer. Co-composting also reduces composting time and can reduce nutrient losses. The research presented in this manuscript, which aimed to determine the impact of biochar addition on N conversion, humification and bacterial community during co-composting of Flammulina velutipes waste with chicken manure, is important from a scientific and practical point of view.

Suggestions for Authors:

1.     Why is the title of the article limited only to Flammulina velutipes waste - a mixture of chicken manure and Flammulina velutipes waste was composted, and why does the title focus on bacterial communities, transformations of C and N compounds and correlations between bacterial communities and quality indicators of humus compounds and nitrogen changes seem to be become the most important results at work.

2.     The proportions are disturbed in the introduction chapter - too little attention is paid to the role of biochar in the composting process

3.     In chapter 2.1 there is no information on the properties of co-composted waste.

4.     In the description of the methods (chapter 2.2): (1) it is imprecisely described how humic and fulvic acids were separated, (2) it is not specified in which extract (HS or HA) spectroscopic properties in the UV-Vis-range were determined, (2) it was not specified what method dissolved organic carbon was determined, (4) it was not specified what methods N-NH4 and N-NO3 were determined.

5.     In the "Results" chapter: (1) the readability of Figures 1 and 2 should be improved, (2) issues related to the indicators of humic substances - the most variable in the composting process and determining the quality of compost, are discussed too generally - they should be discussed in more detail, or in supplementary materials present the results in the form of tables.

6.     In the "discussion" chapter, too little attention was paid to changes in the content and quality of individual groups of humic compounds and determined humification indices, as well as the relationship between the bacterial community and the quality of humic compounds.

7.     The discussion of the results and conclusions did not address how the dose of biochar affects the composting process and the quality of the compost.

Reviewer 2 Report

Comments and Suggestions for Authors

Article title:

“Improve the utilization of Flammulina velutipes waste during biochar-amended composting: Emphasize on bacterial communities".

The work done is certainly of international interest and the format applied is certainly suitable. This manuscript dealt with the topic differently and attractively, and the titles are related to each other. The work is original, of particular interest, and can certainly stimulate research on this topic.

Comments:

Title: Suitable to the topic.

Abstract: Added the design of the experiment; writing the statistical design used and the most important results of the study.

Introduction: Point out the role of biochar in improving the compost process.

Materials and Methods

- Where is the analysis of biochar and its properties?

- Where is the analysis of chicken manure and its properties?

- Where is the analysis of Flammulina velutipes waste?

- Germination Index? Explain the method and mention the name of the seeds used.

- 2.3. DNA extraction and high-throughput sequencing (Add periods for estimation).

Results

- Figure 1. What is the meaning of At? (Temperature, added the below the figure), If possible, clear the Figure.

- 3.1. Effect of biochar on physiochemical and humification during the composting (Point out the

Germination index.

-          Lines: 211 and 233 (Nitrobacter. Bacillus, Itallic).

-          Table 1: added the abbreviation of ACE and PD below the Table.

Discussion

Good writing

Conclusion

Good writing

Reviewer 3 Report

Comments and Suggestions for Authors

The authors proposed a study on using biochar during the composting of Flammulina velutipes waste and chicken manure. The work was well done but some important information must be specified before publication.

• Which biochar was used? How was it produced?

• There is a conflict in describing the method for determining TOC. At L104 it is written that it was determined by an elemental analyzer. This should be total Carbon (TC).

• In general, the “Results” section needs to be rewritten. The results must also be described numerically, reporting the value of the differences and not just the significance. A result is the quantified difference between treatments.

• Some data must also be included in the Conclusions.

• In Fig.1 “At” is room temperature? Rewrite as “Rt”.

Other minor revisions are necessary, highlighted in the attached .pdf file

Comments on the Quality of English Language

Poor English
